# T Regulatory Cells in Inflammatory Bowel Disease—Are They Major Players?

**DOI:** 10.3390/ijms262411944

**Published:** 2025-12-11

**Authors:** Katarzyna Sznurkowska

**Affiliations:** Department of Pediatrics, Pediatric Gastroenterology, Allergology and Nutrition, Medical University of Gdańsk, 80-803 Gdansk, Poland; k.sznurkowska@gumed.edu.pl; Tel.: +48-663-625-861

**Keywords:** IBD, inflammatory bowel disease, T regulatory cells, Tregs infusions, Tregs-based therapies

## Abstract

Inflammatory bowel disease (IBD) is a chronic condition whose pathogenesis is not entirely clear. Impaired immune regulation has been hypothesized as the mechanism responsible for the abnormal response of adoptive immunity to enteric microbial antigens. Regulatory T cells (Tregs) have been regarded as the crucial element of immune regulation, since the discovery that humans lacking Tregs due to mutation of FOXP3 develop autoimmune disorders, including severe bowel inflammation. The existing publications concerning T regulatory cells in human IBD have been reviewed, and current evidence does not clearly indicate quantitative disturbances or functional defects of Tregs in human inflammatory bowel disease. The possible mechanisms explaining immunoregulatory failure in IBD have been summarized. So far, only one clinical trial with Tregs infusion has been completed, and its results do not provide sufficient data on the efficacy or safety of Tregs-based therapies in IBD. It will probably be difficult to implement them in clinical practice in the near future.

## 1. T Regulatory Cell Background

Immune regulation is one of the functions of a normal immune system, which not only has to eliminate invading pathogens but also finally restricts inflammation, avoiding tissue damage. Precise interaction between many cell types is necessary to moderate an effective immune response, thereby maintaining homeostasis and promoting self-tolerance. Among many cell types, regulatory T cells (Tregs) are crucial for maintaining immune balance and protecting against autoimmunity [1]. Their role was first realized with the diagnosis of IPEX syndrome (immune dysregulation, polyendocrinopathy, enteropathy, X-linked). This rare entity, displaying a set of different autoimmune disorders, is caused by mutation of the Foxp3 gene encoding the transcriptional factor that is necessary for Tregs to play their regulatory role [2,3].

Experimental research by Sakaguchi et al. showed that mice undergoing neonatal thymectomy and experiencing a loss of Tregs exhibited multiple-organ autoimmunity, which could be reversed through the adoptive transfer of these cells [4]. From that moment, scientific interest has turned towards the role of Tregs in the pathogenesis of autoimmune disorders. T regulatory cells are CD4+ lymphocytes and can be phenotypically distinguished from other CD4+ subsets by a high cell surface expression of the interleukin-2 receptor alpha chain (IL-2RA), CD25, as well as the aforementioned transcription factor, Forkhead box protein P3 [5,6,7]. Tregs development primarily takes place in the thymus, where in the two-step process of selection the majority of autoreactive T effector cells are depleted, while a small fraction of lymphocyte precursors having “ideal”—not too high, not too low—affinity for interaction with the self-peptide MHC complex convert into T regulatory cells [8].

While other subsets with regulatory activity, like Tr1 cells, Th3 cells, and CD8+ Tregs, have been identified, the classical Tregs characterized by the CD4+CD25highFOXP3+ phenotype appear to demonstrate the strongest immunosuppressive potential among the tested cell populations [9].

The detection of Tregs is based on their characteristic markers by means of cytometry and, also, predominantly in tissues, immunohistochemistry.

Tregs employ various mechanisms to exert their suppressive effects (Figure 1).

Firstly, they secrete immunosuppressive cytokines: interleukin-10 (IL-10), transforming growth factor-β (TGF-β), and IL-35, and also stimulate other cells to produce them [10].

Secondly, Tregs that continuously express cytotoxic T lymphocyte antigen 4 (CTLA-4) inhibit the co-stimulation of effector T cells, which appears to be the most significant mechanism of suppression [10,11].

It has been demonstrated that Tregs facilitate granzyme and perforin-mediated cell death in lymphocytes, primarily targeting CD8+ T cells and natural killer (NK) cells, as well as B lymphocytes [12,13]. Additionally, apoptosis in effector B cells can occur through the programmed death ligand 1 (PD-L1) and programmed cell death receptor (PD-1) pathway [14]. The final mechanism of apoptosis involves cytokine deprivation due to the rapid uptake of IL-2 by CD25+, which is highly expressed on Tregs [15].

Moreover, two mechanisms of Tregs’ impact on dendritic cells (DCs) have been proposed. First, interactions between CTLA4 and CD80/CD86 trigger the release of indoleamine 2,3-dioxygenase (IDO), an enzyme that facilitates the breakdown of tryptophan, which is crucial for lymphocyte proliferation. Second, the binding of lymphocyte-activation gene 3 (LAG3) to MHC class II molecules hinders the maturation and function of dendritic cells (DCs) [10,11,16].

In contrast to effector immune cells Tregs display different metabolic pathways, based mainly on fatty acid oxidation (FAO) and oxidative phosphorylation (OXPHOS), which is crucial for their suppressive function. This reliance on FAO stabilizes FOXP3 expression and enables the resistance of these cells to apoptosis [17].

It has been widely recognized that the population of Tregs is not homogenous, comprising subsets distinguished by varying molecular and cytokine profiles. It was also shown that Tregs originate not only from the thymus, as it was initially believed, but that lymphocytes of the CD4+CD25highFOXP3+ phenotype can also be induced in the periphery from naïve T cells upon antigen exposure, particularly in the presence of TGF-beta and IL-2 [18]. This led to the classification of Tregs into two subsets: natural (thymus-derived, tTregs) and peripheral (adaptive, induced; pTregs) [19].

Peripheral Tregs are predominantly found in barrier tissues and play a key role in preventing local inflammation in response to exogenous antigens [20]. The TCR repertoires of thymus-derived regulatory T cells (tTregs) and peripherally derived regulatory T cells (pTregs) have been shown to differ significantly. In tTregs, the TCR repertoire is primarily biased toward self-recognition, whereas the TCRs found in pTregs are capable of recognizing foreign antigens with high affinity [21]. In accordance with the updated nomenclature, the term iTregs has been designated for cells that are induced in vitro from naïve CD4+ cells [21].

Hence, immune homeostasis mediated by Tregs operates at two levels. Firstly, at a systemic level, during a resting state, it helps maintain tolerance of self-antigens and prevents autoimmunity, primarily through thymic T regulatory cells. Locally, in the context of widespread inflammation, peripheral T regulatory cells mitigate tissue damage caused by immune responses to various environmental factors, such as pathogens, allergens, and trauma. By suppressing different effector cells, Tregs play a beneficial role in protecting against chronic inflammation and allergies [21]. They also contribute to tolerance of fetal antigens, which is essential for maintaining pregnancy [22]. In contrast to these functions, Treg activity can be detrimental in malignancies, as it facilitates immune escape, a phenomenon recognized as one of the hallmarks of cancer [23].

## 2. Tregs in Autoimmune Disorders

Autoimmune diseases comprise a diverse range of poorly understood chronic disorders that impact approximately 5–8% of the population [24]. Although each autoimmune disorder is unique, they all stem from a loss of tolerance to endogenous antigens. This leads to auto-inflammatory events that cause organ destruction via both humoral and cellular immune-mediated mechanisms [25]. Since the first diagnosis of IPEX and the discovery of T regulatory cells, the conception of a decreased number or impaired function of these cells has been hypothesized in the pathogenesis of autoimmune disorders. This led to comprehensive studies of various disease contexts. A quite large number of studies are concerned with the number of Tregs in peripheral blood or, less frequently, at sites of inflammation. Their results are, however, conflicting, with some authors documenting a decrease in circulating Tregs [26,27] and others claiming unchanged [27,28,29] or increased percentages of these cells [30,31]. Furthermore, inconsistencies in the results have been observed even within the same analyzed entities [31,32].

## 3. Tregs in IBD

Inflammatory bowel disease (IBD), which encompasses Crohn’s disease (CD) and ulcerative colitis (UC), is becoming an escalating concern due to its rising incidence and the challenges posed by treatments that are not fully effective and often burdensome [33]. Defective immune regulation has been hypothesized as a mechanism of excessive response to commensal intestinal flora, presently defining inflammatory bowel disease [34]. The discovery of IPEX, which among many autoimmune conditions presents with enteropathy and inflammatory bowel disease, gave the stimuli to regard the role of Tregs in the pathogenesis of IBD. Research has been conducted in parallel on animal models and in humans. Animal-based investigations refer to all types of experimental colitis, which are a great source of knowledge about the biology of these cells [35]. Due to difficulties in obtaining human intestinal Tregs, the investigations in humans are more demanding and less numerous. The great part of human studies refer to the quantification of Tregs in peripheral blood and at sites of inflammation. Others concentrate on the function of these cells and their widely understood properties.

### 3.1. Experimental Studies

Some experimental models of IBD providing valuable insights into mechanisms operative in the development and pathogenesis of these diseases have been used in research on Tregs. They seem to support the role of these cells in IBD. Chemically induced barrier disruption models, such as DSS and TNBS colitis, have demonstrated that the depletion of FOXP3 regulatory T cells in mice leads to an exacerbation of intestinal inflammation [36]. Certain genetic models of spontaneous IBD involve genes affecting Treg function, and treatment with Treg cells can resolve inflammation [37,38]. Other mouse models that use TNF-alpha gene overexpression to induce intestinal inflammation have demonstrated the beneficial effects of Treg stimulation on the progression of murine ileitis [39]. However, the most compelling evidence regarding the role of Treg cells in inflammatory bowel disease (IBD) has come from the T cell transfer model of colitis. In this model, naïve CD4+ T cells, from which Treg cells have been depleted, are adoptively transferred into mice that lack B and T lymphocytes. These effector T cells proliferate and activate in response to bacterial antigens in the intestine, leading to colitis, which can be reversed by the transfer of Treg cells [40,41]. Humanized mouse models, in which mice are repopulated with the human immune system, serve as the most effective tools to study the biology and function of human Tregs in the context of inflammatory bowel disease [42].

Although studies utilizing mouse models support a role for Treg cells in IBD, it has to be emphasized that no single model captures the complexity of human IBD, and the results from experimental studies should never be mechanistically translated to humans. Regardless of whether Tregs disorders were involved in a particular animal model, therapy with these cells leads to a reduction in inflammation. This was the basis for the use of this therapy in humans.

### 3.2. Studies in Humans

The simplest way to investigate the role of Tregs in human IBD is to quantify them in peripheral blood and at sites of inflammation. The proportions of CD4+CD25high or CD4+CD25highFOXP3+ cells in peripheral blood of patients with IBD were compared with the rates of these cells in healthy controls in many studies. The results of these studies were inconclusive, with the majority documenting a decrease in circulating Tregs [43,44,45,46,47,48,49,50,51,52,53,54,55,56] and others claiming unchanged [57,58] or increased percentages of these cells [59,60,61,62] (Table 1).

Discrepancies between the results can be explained by the great heterogeneity of the studied groups. In the majority of the presented publications, the studied groups included patients previously treated, at different stages of the disease [43,44,45,46,48,49,50,51,52,54,55,56,57]. Only a few studies concerned naïve patients, and, notably, four of them demonstrated higher proportions of circulating Tregs in IBD patients compared to healthy controls [47,53,59,60,61,62]. Interestingly, all these studies showing upregulation of Tregs in peripheral blood concerned naïve pediatric patients, which may indicate distinctive mechanisms of IBD pathophysiology in the developmental age [59,60,61,62]. Recently, Duan et al. have attempted to review existing evidence concerning the rates of circulating Tregs in inflammatory bowel disease. All papers included in the analysis referred to adults, and only two out of fifteen concerned naïve IBD patients [63]. The authors concluded that percentages of Tregs in peripheral blood were significantly lower in active UC and CD patients than in healthy controls and inactive IBD patients, while the percentages of Tregs in peripheral blood were comparable between inactive IBD patients and healthy controls [63].

Although most of the existing studies show decreased rates of circulating Tregs, it should be noted that the studied groups are highly heterogeneous in terms of clinical severity, endoscopic stage, treatment, and even disease duration. The negative correlation between the number of Tregs and the disease duration reported in some studies may indicate that the immunoregulatory potential of IBD patients deteriorates over time [64]. In the author’s opinion, only investigations in homogeneous groups of naïve patients, at the onset of the disease, could elucidate the possible role of a Tregs deficit in the pathogenesis of human IBD, both in adults and in children. Additionally, the data concerning bowel mucosae do not confirm the conception of deficiency of Tregs in IBD, showing abundant representation of these cells in inflamed mucosae. The initial report from two decades ago identified CD4+CD25+ regulatory T cells (Tregs) isolated from the intestinal laminae propriae (LPs) of IBD patients. This study showed that these cells are present, express CTLA4, and exhibit in vitro suppressive activity against other T cells similar to that of control groups [65]. All subsequent studies, based on immunohistochemistry, have also demonstrated Tregs accumulation in laminae propriae, which is far more intense in inflamed than uninflamed mucosae [43,45,50,65,66,67,68,69] (Table 2).

Moreover, it has also been shown in the study utilizing cytometry that the rate of Tregs in the bowel is higher than in peripheral blood, which reflects recruitment of immune regulation as a response to inflammation [61]. Despite these results, some authors claimed that the expansion of Tregs is not numerically adequate for the situation.

Maul et al. demonstrated that the quantity of intestinal FOXP3+ cells in inflammatory bowel disease (IBD) is lower compared to that in acute diverticulitis, a condition typically associated with proper immunoregulatory function [43]. In another study, however, no difference was found in Tregs accumulation between IBD and chronic infectious enteritis [70].

Looking at the increasing number of studies showing upregulation of intestinal Tregs, the question arises: Why are they not effective to suppress inflammation? There are some possible answers to this question. The first conception assumes impaired function of Tregs in IBD. The issue has not been fully clarified yet. Some publications demonstrate proper functioning of these cells in inflammatory bowel disease [43,45,65,66,71,72], while others, less numerous, demonstrate distinct results [53,73]. We have to note, however, that assessment of Tregs function is generally carried out using in vitro suppression assays, which quantify the proliferation of conventional T cells when activated in the presence of Treg cells. It is not known how the same cells act in vivo and if their suppressive power is not disturbed by the inflammatory environment in bowel mucosae. Little is known about the impact of other cells or different environmental factors on Treg activity [74,75]. The fact that inflammation persists despite the increased number and proper function of Tregs may indicate that effector cells are resistant to their suppressive activity, as has been described in multiple sclerosis and type 1 diabetes [76,77]. We may also speculate that even if quantitative disturbances of Tregs in bowel mucosae have not been noted, the “quality” of these cells can be invalid, not appropriate to inflammatory status. As is widely known, there are two populations of Tregs: thymus-derived and peripheral, and in the case of the latter, having microbial antigen specificity seems to play a major role in mucosal immunity [19]. As the distinction between these subsets based on different markers such as Helios and Neuropilin-1 proved to be ineffective, we cannot yet determine the origin of Tregs infiltrating intestinal laminae propria and assess their contribution to the regulatory response [78,79]. Moreover, we may suspect that various clones of regulatory T cells are not equally important in preventing IBD. It can be hypothesized that subsets of “appropriate” specificity are not adequately represented in bowel mucosae [80]. Regarding the fact that under inflammatory conditions Tregs can convert to Th17 lymphocytes maintaining FoxP3, we have to realize that detected and quantified Tregs may contain portions of such “fake” Tregs among them. This phenomenon, known as Treg plasticity, refers to their ability to change their phenotype and function in response to specific environmental factors. By the acquisition of molecules characteristic of specific Th cell lineages, they can act as Th17, Th1, or Th2 lymphocytes, contributing to IBD pathogenesis [81].

Future research is necessary to find the answers to these and many other questions. Understanding intestinal Tregs’ biology in settings of inflammation may have the potential to improve therapeutic options for patients with inflammatory bowel disease.

### 3.3. Tregs-Based Therapy in IBD

None of the presently used therapies for IBD are fully effective or safe. They are based on various pharmaceuticals, from the oldest, non-specific medications such as corticosteroids to targeted treatment, including widely used biologic therapies. Regarding the fact that intestinal inflammation is a result of an imbalance between T cell-mediated immune response and immune regulation, the idea of augmenting immune regulatory potential seems to be rational. The theoretical basis for such treatment strategies comes from experimental studies, which for the last 30 years have proved the efficacy of Tregs in curing and preventing IBD in animals. Humanized mouse models repopulated with human immune systems provide new tools to study and develop treatment strategies without putting human lives at risk.

For the last 15 years, many reports on Treg infusion therapies in various autoimmune diseases have been published [82]. The greatest data come from research on type 1 diabetes (T1D), an autoimmune disease with no effective anti-inflammatory treatment, offering replacement therapy as the only therapeutical option. Adoptive Treg cell therapy seems to be an attractive way to restore immune balance and extinguish inflammation in this condition. In this approach, Treg cells are isolated from a patient, enriched, expanded ex vivo, and reinfused. The results of certain completed clinical trials utilizing autologous polyclonal T regulatory cell infusions have shown their safety and efficacy, which was demonstrated by higher c-peptide levels and lower insulin requirements in the studied subjects with T1D [83,84]. Although the majority of clinical trials are designed to treat T1D, some of them have been applied to pemphigus, lupus, autoimmune hepatitis, and, notably, Crohn’s disease [82]. The paper published in 2012 by Desreumaux et al. describes a 12-week open-label, dose-ranging study involving 20 patients with refractory Crohn’s disease [85]. Contrary to the aforementioned studies, the transferred Tregs were selected and cloned to be specific to a dietary antigen (chicken egg ovalbumin), so that antigen-specific activation of the transferred cells could be stimulated in the gastrointestinal tract through an egg-intensive diet. The authors concluded that there was good tolerance of the therapy, although serious adverse events had been observed, and response to the treatment was achieved in 40% of recipients. Unexpectedly, the clinical improvement was inversely correlated with the number of infused cells, and responders had smaller levels of Tregs in the blood compared to non-responders [85]. These results were not promising enough to encourage other researchers to continue the studies. For more than a decade, no other clinical trials utilizing adoptive transfers of Tregs in IBD have been completed; although one clinical trial has recently been registered, it is still in the recruitment phase (TRIBUTE, NCT03185000) [82]. It should be emphasized that we are currently witnessing the dynamic development of non-cell-based therapies tailored to particular mechanisms of IBD pathogenesis, such as anti-interleukin and anti-integrin monoclonal antibodies, Janus kinase (JAK) inhibitors, and sphingosine-1-phosphate receptor (S1PR) modulators, which have been shown to be very effective. They are not associated with the obstacles facing cellular therapies, like the high costs of cellular engineering and ethical issues, as well as the demands of the FDA that they be reproducible, repeatable, and sterile. It does not mean that there is no place for Tregs in the treatment of autoimmune diseases, especially since manufacturing technologies have evolved rapidly during the last decade. Obtaining sufficient numbers of regulatory T cells for therapy is still a major challenge due to their low frequency in peripheral blood. Current methods focus on ex vivo expansion from peripheral blood or in vitro differentiation from naïve T cells, each presenting distinct advantages and limitations [86]. Umbilical cord blood containing a higher proportion of Tregs and lower numbers of T memory cells is another promising source for therapy and has been applied in a few clinical trials. Peripheral blood expansion, currently the most established and clinically used method, is a source of functional and more stable Tregs [86]. The main weakness is the low starting number, necessitating extensive and often complex protocols. The two most common approaches include antibodies bound to beads or cell-based artificial antigen-presenting cells (a APCs). To induce and simultaneously expand Tregs derived from naïve T cells, incubation in the presence of interleukin-2 and TGFβ is employed. These cytokines have also been utilized for the expansion of Tregs isolated from peripheral blood, in conjunction with rapamycin—a cytotoxic T cell inhibitor—applied in both methods [86]. The techniques for isolating and expanding Tregs are continually and successfully being improved to obtain greater quantities of viable Foxp3-positive cells for therapeutic applications.

Recent studies have employed innovative synthetic biology techniques to endow regulatory T cells (Tregs) with unique antigen specificity, including the introduction of specific T cell receptors (TCRs) [87] and chimeric antigen receptors (CARs) [88]. This strategy, which represents the cutting edge in the field, involves the modification of Tregs through the transduction of particular proteins onto their surfaces. The CAR molecule consists of an extracellular antigen-recognition domain, a hinge, a transmembrane region, and an intracellular signaling domain involved in Treg signaling. These CAR-Tregs can migrate to the site of inflammation and bind to tissue-specific autoantigens [86]. The relevant studies have shown not only efficacy in multiple pre-clinical models but also 10–25-fold increased activity compared with polyclonal Treg populations [81]. A lower risk of general immunosuppression is another great benefit of implementing CAR Tregs into cell-based therapies. Taking into account the increasing excellence in cell manufacturing, it cannot be ruled out that such enhanced therapies will be considered in refractory cases of inflammatory bowel disease in the future.

## 4. Conclusions

Although there is considerable interest in regulatory T cells as key players in maintaining intestinal immune balance, there is surprisingly limited evidence linking Treg defects to either type of human inflammatory bowel disease. However, there are many unresolved issues and unanswered questions concerning the pathogenesis of IBD in the context of Tregs function and biology. So far, only one clinical trial applying Tregs therapy in IBD has been completed. Regarding the great progress in pharmacotherapy and the lack of evidence for the effectiveness of Tregs-based therapies in IBD, it will probably be difficult to implement them into clinical practice in the nearest future.

## Figures and Tables

**Figure 1 ijms-26-11944-f001:**
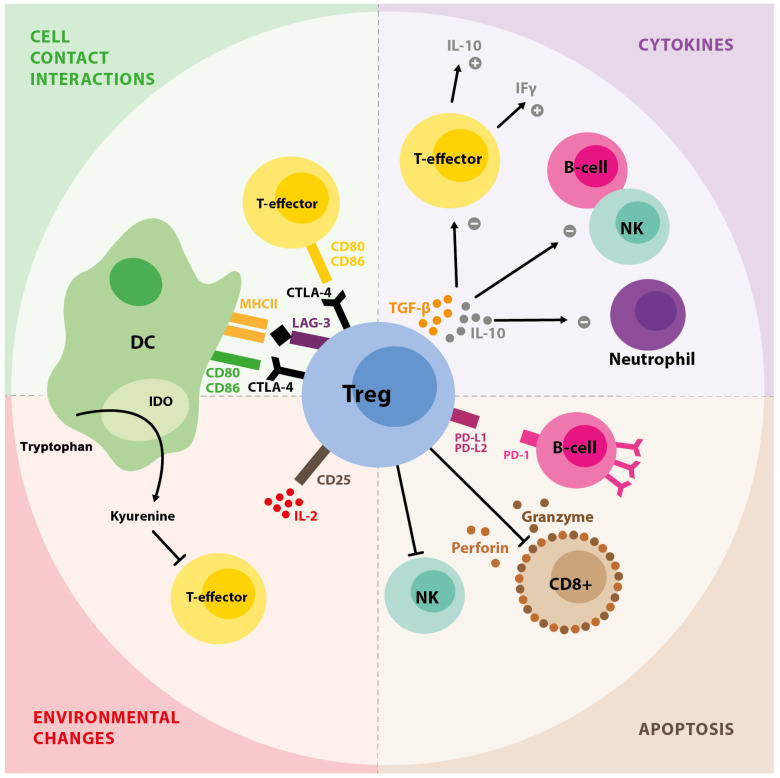
Mechanisms operated by T regulatory cells in inflammation. CTLA-4—cytotoxic T lymphocyte antigen 4, DC—dendritic cell, IDO—indoleamine 2,3-dioxygenase, IL-2—interleukin-2, IL-10—interleukin-10, LAG-3—lymphocyte-activation gene 3, NK—natural killer, PDL-1—programmed death ligand 1, PD-1—programmed death receptor 1, TGF-β—transforming growth factor β.

**Table 1 ijms-26-11944-t001:** Studies reporting on the rates of peripheral blood T regulatory cells in inflammatory bowel disease (in chronological order).

Publication	Disease, Studied Population	Crucial Findings
**Maul J. et al. (2005) [43]**	Adults, N = 46, no information about previous treatment	Decreased number of peripheral blood Tregs in IBD patients, which increase during remission, lower in active disease. Proper function of circulating Tregs.
**Takahashi M. et al. (2006) [44]**	Adults, UC, previously treated	The number of peripheral blood Tregs lower in IBD patients than in HC, the number of Tregs inversely correlated with disease activity.
**Saruta, M. et al. (2007) [45]**	Adults, N = 30, CD, previously treated	The number of peripheral blood Tregs decreased and was inversely correlated with the activity of the disease.
**Yokoyama Y et al. (2007) [46]**	Adults, N = 23, UC, previously treated	Lower number of peripheral blood CD25High+CD4 lymphocytes compared to healthy controls, lower number in active compared to quiescent disease.
**Chamouard P et al. (2009) [47]**	Adults N = 85, CD/UC, naïve patients	The number of peripheral blood Tregs lower in IBD patients than in healthy controls.
**Kamikozuru K et al. (2009) [48]**	Adults = 31, UC, previously treated, non-responders	Decreased levels of Tregs in UC patients not responding to conventional therapy compared to healthy controls.
**La Scaleia et al. (2010) [59]**	Children, N = 43, UC/CD, naïve patients	Increased number of peripheral blood and intestinal Tregs in active disease. Normalization of intestinal Tregs in remission, peripheral Tregs remained increased.
**Di Sabatino et al. (2010) [58]**	Adults, N = 20, CD, previously treated	No significant difference between CD patients and healthy controls. Responders to infliximab presented higher rates of Tregs.
**Boschetti G et al. (2011) [49]**	Adults, N = 25, CU/CD, previously treated	The number of peripheral blood Tregs lower in IBD patients than in healthy controls, but it increased after anti TNFalpha therapy.
**Wang Y et al. (2011) [50]**	Adults, N = 139, UC/CD, previously treated	Decrease in CD4+FOXP3+ Treg cells in peripheral blood and an accumulation of Treg cells in inflamed mucosae.
**Karczewski J et al. (2011) [51]**	Adults, N = 40, UC, previously treated	Decreased frequency of peripheral blood Tregs compared to healthy controls.
**Guidi L et al. (2013) [56]**	Adults N = 32, UC and CD, previously treated	No difference in peripheral blood Tregs between IBD patients and healthy controls.
**Chao K et al. (2014) [52]**	Adults = 46, CD, not treated for 3 mths	Lower rate of peripheral blood Tregs in CD patients.
**Mohammadnia-Afrouzi M et al. (2015) [53]**	Adults, N = 32, UC, naïve patients	Decreased rate of peripheral blood Tregs in UC patients, inversely correlated with disease activity.
**Sznurkowska K et al. (2016) [60]**	Children; N = 24; UC/CD, UC, and CD; naïve patients	Increased number of peripheral blood and intestinal Tregs in naïve IBD patients.
**Gong Y et al. (2016) [54]**	Adults, N = 90, UC, treatment history not described	Decreased rate of peripheral blood Tregs in UC patients.
**Khalili A et al. (2018) [55]**	Adults, N = 23, CD	Decreased rate of peripheral blood Tregs in CD patients, proper Tregs, function.
**Sznurkowska K et al. (2020) [61]**	Children, N = 15, UC and CD, naïve patients	Increased number of intestinal and circulating Tregs, the rate of Tregs in bowel mucosae higher than in the blood.
**Vitale A et al. (2020) [62]**	Children, N = 35, UC/CD, naïve patients	Increased number of circulating and intestinal Tregs in naïve IBD patients.
**Lin QR et al. (2020) [56]**	Adults, N = 32, CD, previously treated	Decreased rate of peripheral blood g Tregs in CD patients before Infliximab therapy.

**Table 2 ijms-26-11944-t002:** Publications concerning mucosal (intestinal) T regulatory cells in inflammatory bowel disease (in chronological order).

Publication	Disease, Studied Population	Crucial Findings, Method Used
**Makita et al. (2004) [65]**	Adults, N = 49, CU/CD, previously treated	Increased number of intestinal Tregs compared to healthy controls, proper function of Tregs isolated from colonic biopsies. Cytometry.
**Maul J. et al. (2005) [43]**	Adults, N = 46 (24 active disease), no information about previous treatment	Increased number of intestinal Tregs compared to healthy controls, but lower than in acute diverticulitis. Immunochemistry.
**Holmen N et al. (2006) [66]**	Adults, N = 39, UC, no information about previous treatment	Increased amounts of Tregs in inflamed mucosae, correlation with disease activity, proper function. Cytometry.
**Saruta, M. et al. (2007) [45]**	Adults, N = 30, CD previously treated	Inflamed mucosae enriched in Tregs, proper function ex vivo. Cytometry.
**Li et al. (2010) [67]**	Adults, N = 40, CU/CD, previously treated	Increased density of Tregs in inflamed mucosae. Immunochemistry.
**Reikvam, D et al. (2011) [68]**	Adults, N = 12, children, N = 14, CD naïve patients	Increased density of Tregs in inflamed mucosae. The number of Tregs higher in children than in adults. Immunochemistry.
**Wang Y et al. (2011) [50]**	Adults, N = 139, UC/CD, previously treated	Accumulation of Treg cells in inflamed mucosae. Immunochemistry.
**K Sznurkowska et al. (2017) [69]**	Children, N = 66, UC/CD Naïve patients	Increased density of Tregs in inflamed mucosae compared to IBS patients. Immunochemistry.

## Data Availability

No new data were created or analyzed in this study. Data sharing is not applicable to this article.

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
