# Peer review of "T Regulatory Cells in Inflammatory Bowel Disease—Are They Major Players?"

_ijms, 2025, doi:10.3390/ijms262411944_

Round 1
Reviewer 1 Report
Comments and Suggestions for Authors
I go through the review “T Regulatory Cells in Inflammatory Bowel Disease- Are They Major Players?” where the author reviewed existing evidence on the role of regulatory T cells (Tregs) in the pathogenesis of human inflammatory bowel disease (IBD). The investigation focused on whether quantitative disturbances or functional defects in Tregs are a key mechanism in IBD. The author concluded that current data does not clearly support a major Treg deficiency in human IBD, explored potential reasons for immunoregulatory failure, and assessed the current state and future challenges of Treg-based cellular therapies for IBD. I found the review very interesting and I do support it. However I suggest the author to significantly revise and add these few suggestions.
- A section detailing the thymic development of natural Tregs (tTregs) and the peripheral induction of adaptive Tregs (pTregs) would add foundational biological context.
- Discussing the challenges of obtaining sufficient numbers of Tregs for therapy, comparing sources like peripheral blood expansion versus in vitro differentiation from naïve T cells.
- Exploring the cutting-edge field of engineering Tregs, such as with Chimeric Antigen Receptors (CAR-Tregs), would make the review more forward looking.
- Detailing the methods and hurdles in expanding Tregs in the laboratory while maintaining their stable regulatory phenotype.
- Including how the metabolic state of Tregs (e.g., reliance on fatty acid oxidation) is crucial for their function and could be a relevant topic.
- Reviewing the evolution of methods to isolate pure Treg populations (e.g., CD4+CD25+CD127low) for research and therapy.
- More topography should be added to the review manuscript. Such as Treg cells in tissue functioning, Treg structure, shape etc. these are just few example.
- I also suggest the author to add a sectiona about what the other roles of Tregs in body functioning other than IBD.
- Please revise the title to be scientific and logical now a question.
- The keywords need to be ascending alphabetic order.
- The tables format need to be corrected, such as lines spaces should be 1 and remove space before and after paragraph. The full detail of publication should be removed, only citation is enough. Publication column should be moved to the right side. Top row and left column should be bold.
- Interlink all the citations to corresponding references. Also interlink all the mention of fig. and table in text to corresponding figure and tables.
- The review paper contains 42% plagiarism which is too high, I suggest the authors to make correction to this issue and before submitting the revised version make sure the plagiarism is below 20%.
Overall, the review is good one and I do support this however the current state of manuscript is not in a position to be published. I suggest the author to significantly expand and revise the review paper as per above suggestions.
Author Response
Dear Reviewer
I would like to thank you for your kind, an in-depth review of this manuscript and for the very valuable comments. I have read the comments carefully and revised the manuscript based on included suggestions. Changes to the manuscript have been highlighted in red. Replies to all the remarks are enclosed in separate file.
I am very grateful for the opportunity to submit a revised manuscript and I am looking forward to your next correspondence.
Sincerely
Katarzyna Sznurkowska

Reviewer 2 Report
Comments and Suggestions for Authors
This manuscript presents a thorough and well-referenced review of the role of regulatory T cells (Tregs) in the pathogenesis and potential treatment of inflammatory bowel disease (IBD). The author critically examines both experimental and clinical evidence, highlighting the complexity and ambiguity surrounding Treg role in IBD. The manuscript is generally well-written, scientifically sound, and well-supported by relevant literature.
Major comments
The reported similarity index of 42% is unacceptably high for a scholarly manuscript. This level of overlap raises serious concerns regarding originality and proper citation practices. I strongly recommend that the author thoroughly revises the manuscript to eliminate redundant or improperly attributed content and ensure compliance with academic integrity standards.
Minor comments
- Please ensure that all acronyms are defined upon first use. For example, “NK” on line 65 should be introduced as “natural killer (NK) cells.
- The distinction between Table 1 and Table 2 could be made clearer through more precise titles. Table 1 summarizes studies on Tregs) in peripheral blood, while Table 2 focuses on Tregs in intestinal mucosa. I suggest revising the titles to explicitly reflect the biological compartment analyzed (e.g., “Peripheral Blood Tregs in IBD” and “Mucosal Tregs in IBD”) to help readers quickly understand the context and avoid confusion.
- The manuscript could benefit from a more detailed exploration of recent findings on Treg plasticity, especially their potential conversion to Th17 cells under inflammatory conditions.
- The section on Treg-based therapy is brief. Expanding on current clinical trials, challenges in cell therapy, and future directions would enhance the translational impact.
Author Response
Dear Reviewer
I would like to thank you for your kind and careful review of this manuscript and for the very valuable comments. I have read the comments carefully and revised the manuscript based on included suggestions. Changes to the manuscript have been highlighted in red. Replies to all the remarks are enclosed in separate file.
I am very grateful for the opportunity to submit a revised manuscript and I am looking forward to your next correspondence.
Sincerely
Katarzyna Sznurkowska

Reviewer 3 Report
Comments and Suggestions for Authors
The manuscript aims to highlight the updated comprehensive role of Treg in IBD. The manuscript is clear, well structured, and the gap was identified. The cited references are updated and relevant. The figure and tables properly show the data and understandable. the statements and conclusion drawn coherent and supported by the listed citations.
Minor comments:
1- page 1 line 24 and 25:the author didn't mention IBD at all under this part, so I suggest to change his subtitle to T reg function.
2- page 3 line 64 and 65, I suggest listing the ro.e of PD-1 and PD-1L role with appropriate citation.
Author Response

(The authors gave the same response as above.)

Round 2
Reviewer 1 Report
Comments and Suggestions for Authors
Please remove the plagiarism upto below 20%.
Author Response
Dear Reviewer ,
I am grateful for your kind review . I have done everything to achieve the desired level of plagiarism, but it seems impossible to me. I am impressed by the amount, but also by the quality of phrases , which were indicated as plagiarism by the program used by MDPI.
Let us take such phrases, which by no means can be replaced by any other expression:
T Regulatory Cells in Inflammatory Bowel Disease- Title!
diagnosis of IPEX syndrome (immune dysregulation, polyendocrinopathy, enteropathy, X-linked- just the definition of the syndrome - line
IBD; inflammatory bowel disease ;T regulatory cells; Tregs- key words,
high cell surface expression of the interleukin-2 receptor alpha chain (IL-2RA), CD25,- l42-explanation of,,CD25 high,,meaning
affinity of interaction with the self-peptide MHC complex-
PDL-1-programmed death ligand -1, PD-1, programmed death receptor 1,TGF-β - transforming growth factor
secrete immunosuppressive cytokines: interleukin-10 (IL-10), transforming growth factor-β (TGF-β), and IL-35,
cytotoxic T lymphocyte antigen 4 (CTLA-4)
CD8+ T cells and natural killer (NK) cells.
And many other examples- mentioning them is too much time consuming.
Even the heading: Int. J. Mol. Sci. 2025, 26, x FOR PEER REVIEW- has been marked as a plagiarism on each of 13 pages of the manuscript
Another category includes sentences, concerning so widely known facts , that do not even need to be checked or found anywhere. I have simply known them, because I have read about them so many times – they simply and directly came from my mind. I think theynprobably appear in many presentations, textbooks, publications.
Here are the examples
IBD is a chronic condition of not entirely clear pathogenesis.
Impaired immune regulation has been hypothesized as the mechanism responsible for the abnormal response of the adoptive immunity to enteric microbial antigens – this is just definition
Autoimmune diseases represent a heterogeneous group of poorly understood long-term disorders that affect approximately 5-8% of the population
Regulatory T cells (Tregs) have been regarded as the crucial element of immune regulation, since the discovery that humans lacking Tregs due to mutation of FOXP3 develop autoimmune disorders including severe bowel inflammation lines .
In this approach, Treg cells are isolated from a patient, enriched, expanded ex vivo, and reinfused
Notably majority of the detected plagiarism 15/31% comes from not specified publications , but from the so- called internet.
Another category of plagiarism refers to presentation of the results of the quoted research with the most representative example:
Decrease of CD4+FOXP3+ Treg cells in peripheral blood and an accumulation of Treg cells in inflamed mucosa, - it appeared in many places as many publications have shown similar results
Or
The authors concluded that percentages of Tregs in peripheral blood were significantly lower in active UC and CD patients than in healthy controls and inactive IBD patients, while the percentages of Tregs in peripheral blood were comparable between inactive IBD patients and healthy controls- this was the conclusion of the study
or
demonstrated by higher c-peptide levels and lower insulin requirements in the studied subjects
According to the definition of plagiarism (Plagiarism is presenting someone else's words, ideas, or work as your own without proper acknowledgment, essentially stealing intellectual property and failing to credit the original creator,) none of the displayed above match this definition, especially that all the sentences refer to existing publications.
Summing up, I have once again revised the manuscript critically, trying to avoid long, original sequences of words which have been probably cited ( it is not a sin) , and tried to retell them in another way ,sometimes not so excellent. I do not find this activity beneficial for the manuscript. The changed text has been marked in green
I have attached the text with the marked plagiarism and I would kindly request to analyze it to see the high imperfection of AI plagiarism detection or even its failure.
In my opinion this so-called, plagiarism rate , does not reflect real plagiarism at all. It is also worth noting, that the character of this manuscript- review , contrary to ,,original paper”, is based on what is known , and the ,, originality” demands are obviously lower than in the original papers
Once again I count on your understanding. I am looking forward to your next correspondence.
Sincerely
Katarzyna Sznurkowska

Reviewer 2 Report
Comments and Suggestions for Authors
The manuscript has improved significantly. Regarding my earlier comment, I did not dispute the term “Tregs”; rather, I was suggesting that the titles of Table 1 and Table 2 should explicitly indicate the biological compartment analyzed. For example:
- Table 1: Peripheral Blood Tregs in IBD
- Table 2: Mucosal Tregs in IBD
This will help readers quickly understand the context and avoid confusion.
Author Response
Dear Reviewer ,
I am grateful for your kind review . I have changed the tables' titles accordingly.
Sincerely
Katarzyna Sznurkowska